# Larval Morphological Adaptations of *Leiodes cinnamomea* (Panzer, 1793) (Coleoptera: Leiodidae: Leiodinae)—Obligatory Feeder of *Tuber* Species

**DOI:** 10.3390/insects13030249

**Published:** 2022-02-28

**Authors:** Aleksandra Kilian, Marcin Kadej, Jonathan Cooter, Deborah J. Harvey

**Affiliations:** 1Department of Biology, Evolution and Conservation of Invertebrates, University of Wrocław, Przybyszewskiego 65, 51-148 Wrocław, Poland; 2Hope Entomological Collections, Department of Life Sciences, Oxford University Museum of Natural History, Parks Road, Oxford OX1 3PW, UK; jon.athanc6096@gmail.com; 3Department of Biological Sciences, Royal Holloway University of London, Egham TW20 0EX, UK; d.harvey@rhul.ac.uk

**Keywords:** morphology, larva, biology, subterranean mycophagy, *Leiodes cinnamomea*, *Tuber aestivum*, pseudomycangia, mosaic strategy

## Abstract

**Simple Summary:**

The family Leiodidae include beetles adaptated to many various habitats and foods ranging from saprophagous by obligatory mycophagists to true ectoparasites. The phylogeny of the family is complex and still requires research. Larval stages have different characters to adults and provide a lot of phylogentically valuable data. However, there are still gaps in our knowledge about immatures of the family, amongst others, in the tribe Leiodini, which is connected with endogean fungi. We provide the first detailed study of morphology of larval stages of *Leiodes cinnamomea*, which is an obligatory feeder and a pest of the economically valuable truffle. Our study found that larvae had unique morphological characteristics. This uniqueness, with larvae of all stages having a specific arching behaviour, is probably more common in this genus and is connected with endogean mycophagy and with other morphological characteristics. We present our hypothesis, based on our observation of the behaviour and morphology, that larvae of *L. cinnamomea* use a mosaic strategy for beetles living in ephemeral and in solid fungi. The study also showed that many other morphological features, such as chaetotaxy and length ratios of other parts of the body, confirmed the phylogenetic relationship that puts Leiodini close to Scotocryptini.

**Abstract:**

Detailed redescription of morphology for three larval instars of truffle-feeder *Leiodes cinnamomea*, documented on scanning micrographs and illustrations, is provided. Systematic context, observations on biology and unique characteristics of larvae of the only well-described representative of Leiodini are discussed. Exoskeletal invaginations (pseudomycangia), granulation on the head and the median longitudinal row of sclerotized plates on tergites VII–IX—the unique structures among leiodid larvae—were confirmed, described and documented. A mosaic strategy for beetles living in ephemeral and in solid fungi is discussed.

## 1. Introduction

*Leiodes cinnamomea* (Panzer, 1793) is a member of one of the most species-rich genera of Leiodinae, numbering about 200 species [1,2,3]. The genus has Holarctic distribution [4,5,6,7], with some species also found in the northern part of Neotropical and in the Oriental region [3,8,9,10,11,12], as well as some in South Africa [13,14]. It is thought that all members of the genus, as well as other genera of Leiodini, feed on subterranean fungi and, as with most Leiodinae, are obligate mycophages [15,16]. Many species are thought to develop in mycelia in the roots of grasses and rhizoids of mosses [13].

*L. cinnamomea* is one of the most specialized fungivores [17] and the most economically significant member of Leiodidae. The species is thought to be associated with almost all species of the genus *Tuber* Mich. ex Fr. across Europe [17,18,19]. As its common name, ‘truffle beetle’, indicates, it damages the fruiting bodies of the ‘black truffle’ *Tuber melanosporum* Vittadini [19,20], the ‘white truffle’ *Tuber magnatum* Pico [18], the ‘autumn truffle’ *Tuber aestivum* Vittadini [21,22,23,24], the ‘hollowed truffle’ *Tuber excavatum* Vittadini [24] and the ‘Burgundy truffle’ *Tuber uncinatum* Chatin, 1892, a synonym of *T. aestivum*, but distinguished from it in culinary circles. It has also been recorded in association with *T. brumale* Vittadini [19,20,22]. Previous studies show that other leiodid species are also connected with *T. aestivum*, *T. excavatum* and *T. rufum* [23,24].

Although the species *L. cinnamomea* is a pest of the most economically valuable fungi, its biology was only recently studied by Arzone [18,19]. She determined the life cycle, behaviour and ecology, embryonic development, eggs and larval stages, as well as pupa. *L. cinnamomea* has three larval instars, typical of most Leiodidae [2,18], with the first and the second instar lasting 8 days. The last instar has a period of 5–8 days of trophic activity and 5–6 days for gut emptying and formation of the pupal cell, where it will spend about 8 months on larval diapause. The life cycle of this univoltine species takes place in the winter and early spring months, with adults and larvae living in low temperatures, coinciding with the life cycle of truffles. Both adults and larvae are photophobic.

Arzone [19] described the morphology of the first instar larvae; until recently, this remained the only detailed work on larval morphology for the 429 known species of Leiodini. A short note by [25] concerning *Leiodes rufipennis* (Paykull, 1798) was more concerned with habitat, whereas older descriptions have only historical importance [26,27]. The only study about this interesting species in the present century concerned the olfactory orientation of the adult beetle, which appeared to not be attracted by ripe truffle odours; rather, male beetles are attracted by the presence of females in a truffle [20]. Hochberg et al. [20] hypothesized that truffle volatiles attract insects at an early stage and that, just before decomposition, the truffle becomes attractive to mammals. The chemical changes involved in this process and the attractiveness of a truffle to *Leiodes* are not fully understood. It is likely that the dissemination and the germination of spores involves beetles, but the actual process is still unknown [18,28]. However, the correlation of the life cycle of *Colenis immunda* (Sturm, 1807), *L. cinnamomea* and *Leiodes oblonga* (Erichson, 1845) with fructification of *T. aestivum* has been confirmed [24]; it is believed that the correlation between the appearance of *L. cinnamomea* and truffles may be vital in understanding the relationship between the beetle and fungi. The woodland in southern England, from which the adult beetles and truffles for our research were collected, was planted in 1986 on land that had been in arable use for generations; truffles (*T. aestivum*) were noted about three years later, with *L. cinnamomea* being observed two years after arrival of truffles (pers. obs. JC).

Considering the economic importance of truffles, studying this univoltine, cryophilic, photophobic beetle species is worthwhile because of the many unknowns surrounding its biology, ecology and influence on the distribution of the fungi. Moreover, detailed comparative larval morphology shows unique characters of the larvae of this species and will inform understanding of the systematic context of *L. cinnamomea* within Leiodidae, especially since it is currently the only detailed larval description of Leiodini.

## 2. Materials and Methods

### 2.1. Materials

England: Wiltshire, ex. adults, 23.11.2014, from *Tuber aestivum,* leg. J. Cooter; 16 females, 9 males, 5.01.2015, from *Tuber aestivum,* leg. J. Cooter; 8 females, 3 males, 26.09.2015, from *Tuber aestivum,* leg. J. Cooter.

20 larval specimens were cultured in the laboratory in Royal Holloway University of London, UK and University of Wrocław, Poland.

Poland: near Sękocin, 1 larva, 2 adults, 31.08.2015, from *Tuber excavatum*, leg. A. Rosa-Gruszecka; near Sękocin, 6 larvae, 04.09.2015, from *Tuber excavatum*, leg. A. Rosa-Gruszecka; near Pińczów, 1 larva, 03.10.2015, from *Tuber excavatum*, leg. A. Rosa-Gruszecka; near Sękocin, 2 larvae, 17.11.2015, from *Tuber excavatum*, leg. A. Rosa-Gruszecka.

### 2.2. Breeding

Individuals of *L. cinnamomea* were bred in plastic or metallic boxes (ca. 10 cm/10 cm/5 cm) filled with alkali soil (a mixture of soil from the collection locality and soil from other sources with natural chalk added) and supplied with mature *T. aestivum* collected with the *L. cinnamomea* adults. The boxes were kept at a constant temperature of 10 °C and were checked frequently (every 1–2 days) in order to capture the key moments in the life cycle of the beetles. Larvae were kept together with adults and fed with the same medium. Eggs, larvae of all instars, Acarina and larvae of Diptera were collected and preserved in 75% ethanol; nematodes were preserved into 99% ethanol.

### 2.3. Preparation

Larvae and eggs were preserved in 75% ethanol. Two specimens of the first instar, two of the second instar and one of the third instar were macerated in warm 10% KOH and washed in distilled water. The heads of some specimens were separated before maceration and mouthparts were separated and dissected. All parts of the body were mounted in glycerol-gelatin.

### 2.4. Light Photography and Image Processing

Drawings, measurements and photographs of details were made at magnification up to 1000× using a Nikon^®^ Eclipse 80i Phase Contrast microscope (Tokyo, Japan) with a drawing tube. Photographs of morphological details were made by a Nikon Coolpix^®^ 4500 digital camera (Tokyo, Japan). Photographs of habitus were made under an Olympus SZ2^®^ stereo microscope with an Olympus SC30^®^ digital camera (Hamburg, Germany). Image stacks were processed using Combine ZP^®^ (Derby, UK).

### 2.5. SEM Preparation

One example of the third, second and first instar larva was dehydrated in a graded series of ethanol baths, dried using hexamethyldisilazane, mounted on an SEM stub with a carbon tab, coated with gold using Leica EM ACE250 (Wetzlar, Germany) and examined with a ZEISS EVO^®^LS15 scanning electron microscope (Oberkochen, Germany) at magnification up to 20.000×.

### 2.6. Terminology and Measurements

Terminology used for the chaetotaxy follows systems proposed by Ashe and Watrous [29], Wheeler [30] and Kilian [31,32]. Table 1 provides average measurements of three larval instars. 

In addition, the following indices are used: AI/AII/AIII/SA, LPI/LPII, MPI/MPII/MPIII, URI/URII/URS.

## 3. Results

### 3.1. Biology

Eggs (Figure 1A) were laid separately, directly in the ground or on the surface of truffle, at random locations. Different larval stages were simultaneously found on, inside and around the truffle body and in the soil. Larvae moved very quickly and used pygopods for moving and for defense (Figure 1B–F). They bent up their abdomen, straightened their head and moved large mandibles. Mature larvae (Figure 2A,B) made cells for diapause and pupation [33] in the fungi using corridors in the hymenium (Figure 2C), which were closed by a cup formed from the hymenium chewed by the mandibles (Figure 2D) or in the soil (Figure 2E,F). A complete life cycle from an egg to adult, in the laboratory, at 10 °C, is not known, as we did not observe pupation.

We also observed other invertebrates: numerous representatives of Acarina *Zerconopsis remiger* (Kramer, 1876) (Figure 3A), some dipteran larvae *Bradysia smithae* Menzel & Heller, 2005 [34] and numerous nematodes (Figure 3B) on the truffle body. However, we did not observe interactions between *L. cinnamomea* and these organisms, except one incidence of nematodes in the cranium of a dead larva of *L. cinnamomea*.

**Figure 1 insects-13-00249-f001:**
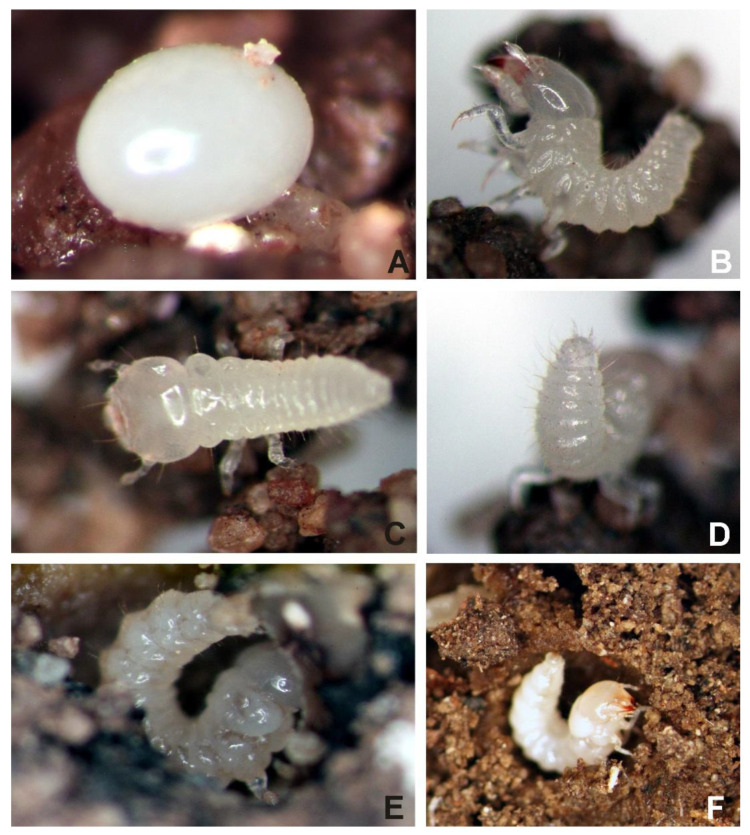
*Leiodes cinnamomea*. (**A**) Egg. (**B**) Larva of the first instar with abdomen arching up. Habitus of larva of the second instar feeding hymenium of *Tuber aestivum*. (**C**,**D**) Habitus od larva of the second instar using pygopode for moving. (**E**,**F**) Larva in dorsal view with abdomen arching up making cocoon.

**Figure 2 insects-13-00249-f002:**
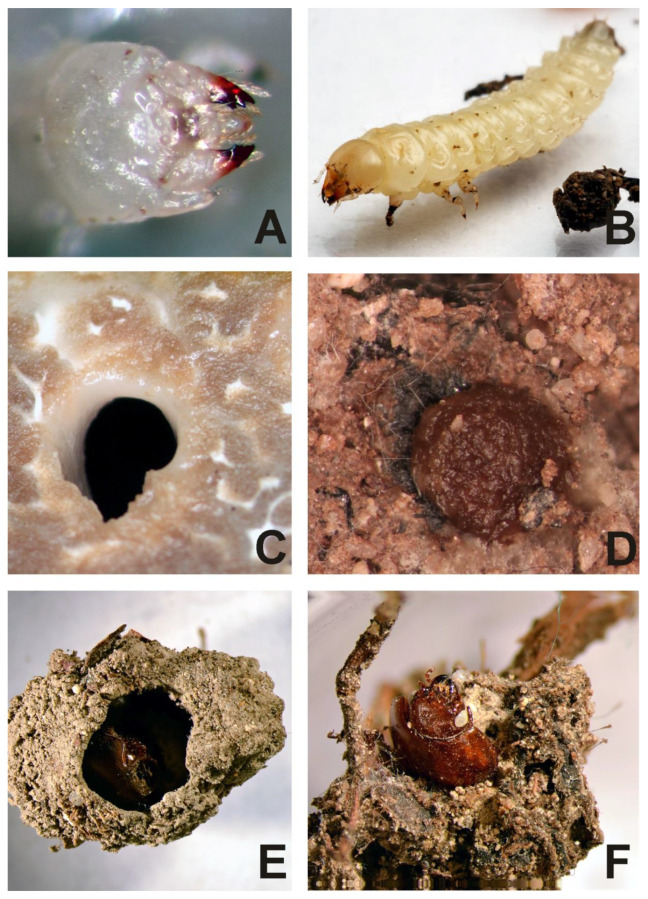
*Leiodes cinnamomea*. (**A**) Head of the second instar. (**B**) Habitus of larva of the third instar. (**C**) Entrance of corridor made by beetles in *Tuber aestivum*. (**D**) The cup of cocoon made by larva from chewing hymenium of truffle. (**E**,**F**) Adult beetle in cocoon formed from soil.

**Figure 3 insects-13-00249-f003:**
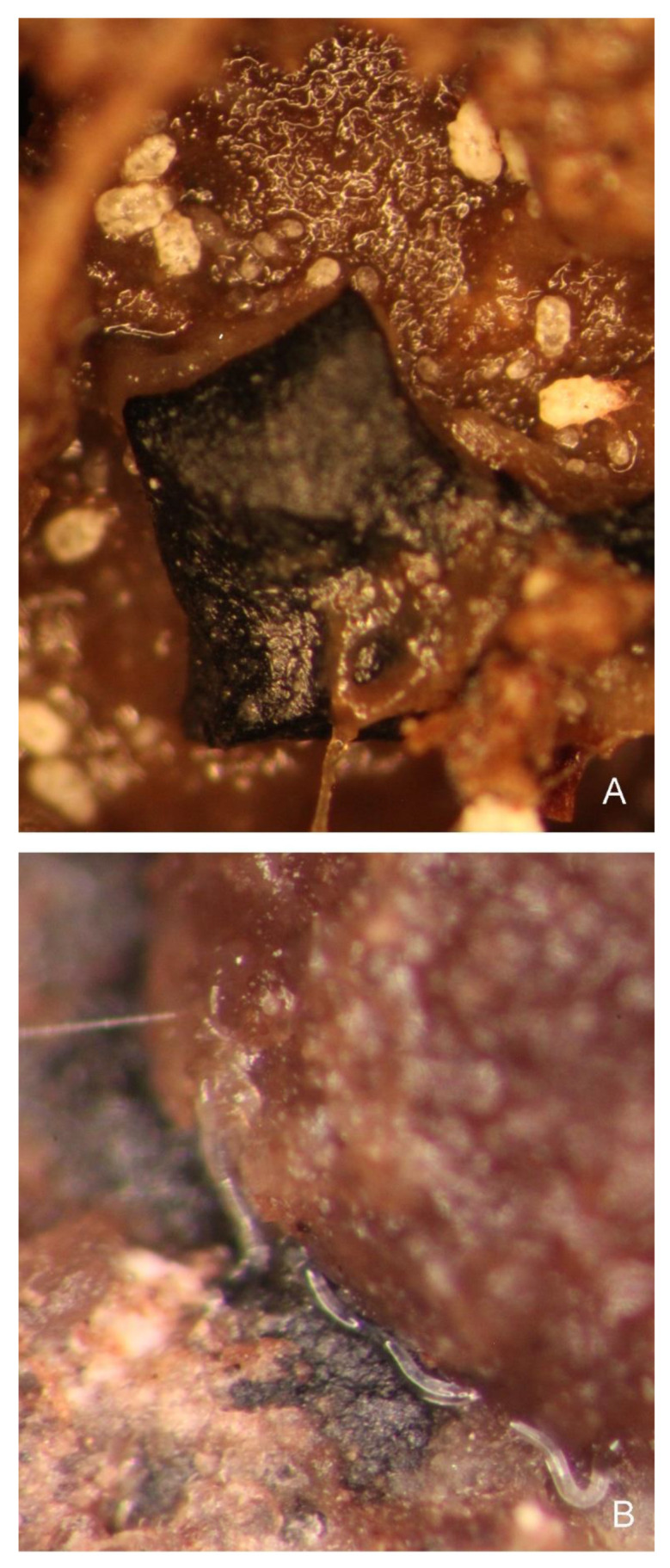
*Tuber aestivum*. (**A**) Acarina *Zerconopsis remiger* on surface of fungi. (**B**) Nematodes on surface of truffle near a cup of cocoon made by larva of *Leiodes cinnamomea*.

### 3.2. Description of the Immature Stages of Leiodes cinnamomea


**Egg (**
Figure 1
**A)**


Colour white, shape ovoid, surface smooth, average length: 1.16 mm (1 mm–1.28 mm), width: 1.1 mm.

**Body. (Figure 1B–F, Figure 2B, Figure 4A–H, Figure 5A–I, Figure 6A–I, Figure 7A–F, Figure 8A–I, Figure 9A–F, Figure 10A–F, Figure 11A–F, Figure 12A–F, Figure 13A–F, Figure 14A–F, Figure 15A–F, Figure 16A–C, Figure 17A,B, Figure 18A–F and Figure 19A–I; Table 1)**.

Total body length: *Instar I*: 4.5 mm. *Instar II*: 6.15 mm. *Instar III*: 10 mm. White, with thin smooth cuticle, with large head (proportionally to the body). Head with very small antennae and without stemmata and characteristic granulation on frontal part and paired sclerotized invaginations below antennal. Large asymmetric mandibles with reddish-brown apices (Figure 2A,B). Larvae in all instars densely setose with small setae on ventral side of abdomen. Setae simple, brownish. Each segment of abdomen divided into tergite (mediotergite) and laterotergite with annular spiracle. Tergites (mediotergites) undivided medially, without carina, with paired reservoirs present and with two pairs of long posterior setae. Sternites with three pairs of long, posterior setae and numerous minute setae on mediosternites (only in instar III) and group of minute setae on laterosternites, which are less isolated than laterotergites. Asperities arranged into transverse short rows on mediotergites and mediosternites. Abdominal mediotergites VII–IX with medial row of sclerotized plates. Very small two-segmented urogomphi on abdominal segment IX. Brownish tarsunguli of legs.

*Head*. Prognathous. Cranium wider than long. Stemmata absent. Epicranial stem and sutures absent. Regular granulation of frontal area (Figure 4A, Figure 5B, Figure 6A, Figure 7E,F, Figure 9A and Figure 13B). Paired sclerotized invaginations dorsally, prolonged from dorsal mandibular articulations (Figure 7C,D and Figure 8D–F). *Instar I*. (Figure 4A) HW/HL: 1.15. HW: 0.81–0.88 (av. 0.84 mm); HL: 0.69–0.78 (av. 0.73 mm). Chaetotaxy of dorsal side: Row Da with four setae (Da*, Da**, Da1, Da2); row Db with two setae (Db1, Db2); row Dc with two setae (Dc, Dc2); row Dd with two setae (Dd1, Dd2); row De with two setae (De1, De2); numerous (ca. 15) minute setae on each side of head; two pairs of campaniform sensilla. Ventral side with six–seven setae on each side of head, microsculpture absent. *Instar II.* (Figure 5B and Figure 13A–C). HW/HL: 1.22. HW: 1.25–1.42 (av. 1.34 mm); HL: 0.90–1.28 (av. 1.1 mm). Chaetotaxy of dorsal side same as in instar I. Ventrally with 10 setae on each side of head. *Instar III.* (Figure 6A, Figure 7A–F, Figure 8A,D–F and Figure 9A,B). HW/HL: 1.34. HW: 1.60 mm; HL: 1.00–1.44 (av. 1.19 mm). Chaetotaxy of dorsal side same as in instar I and II except more numerous minute setae (ca. 35 on each side of head). Ventrally ca. 10–15 setae on each side of head, microsculpture absent.

**Figure 4 insects-13-00249-f004:**
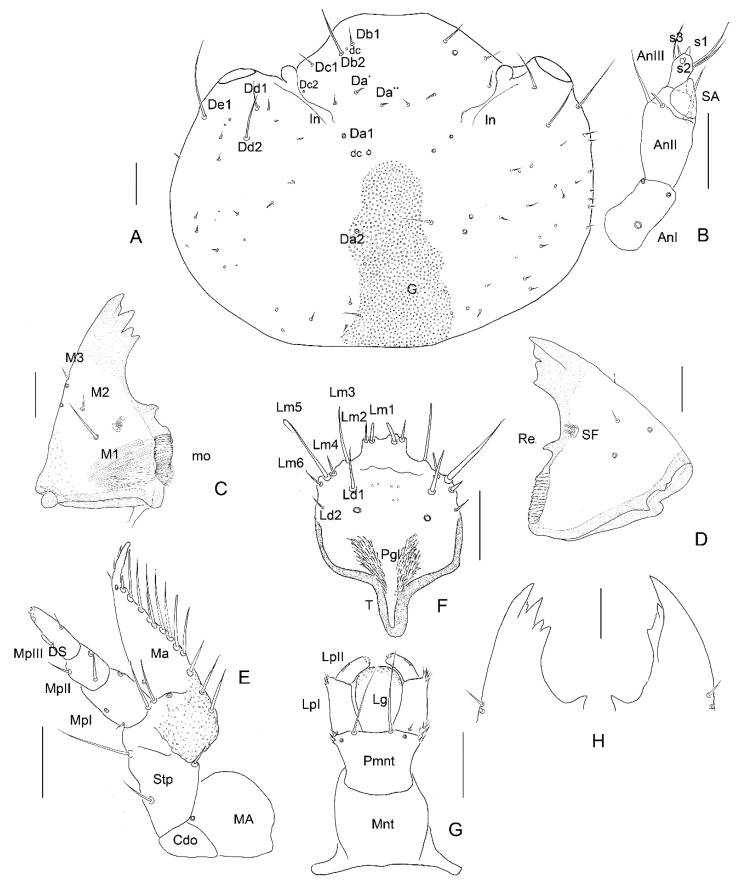
*Leiodes cinnamomea*. Details of larval morphology of instar I. (**A**) Dorsal side of head. (**B**) Antenna. (**C**) Left mandible, ventral view. (**D**) Right mandible, ventral view. (**E**) Maxilla, ventral view. (**F**) Labrum, ventral view. (**G**) Labium, ventral view. (**H**) Apices of mandibles. Abbreviations: AnI, AnII, AnIII—antennomere I, II, III; SA—sensory appendage: Cdo—cardo; DS—digitiform sensillum; G—granulation; In—invagination; LpI, Lp II—labial palpi segment I, II; Lg—ligula; MA—maxillary area; Ma—mala; Mnt—mentum; Mo—mola; MpI, MpII, MpIII—maxillary palpi segment I, II, III; P—pocket; Pgl—paraglossae; Pmnt—prementum; Re—retinaculum; SF—setal field; Stp—stipes. The other abbreviations refer to the terminology of setae. Scale bars = 0.1 mm.

**Figure 5 insects-13-00249-f005:**
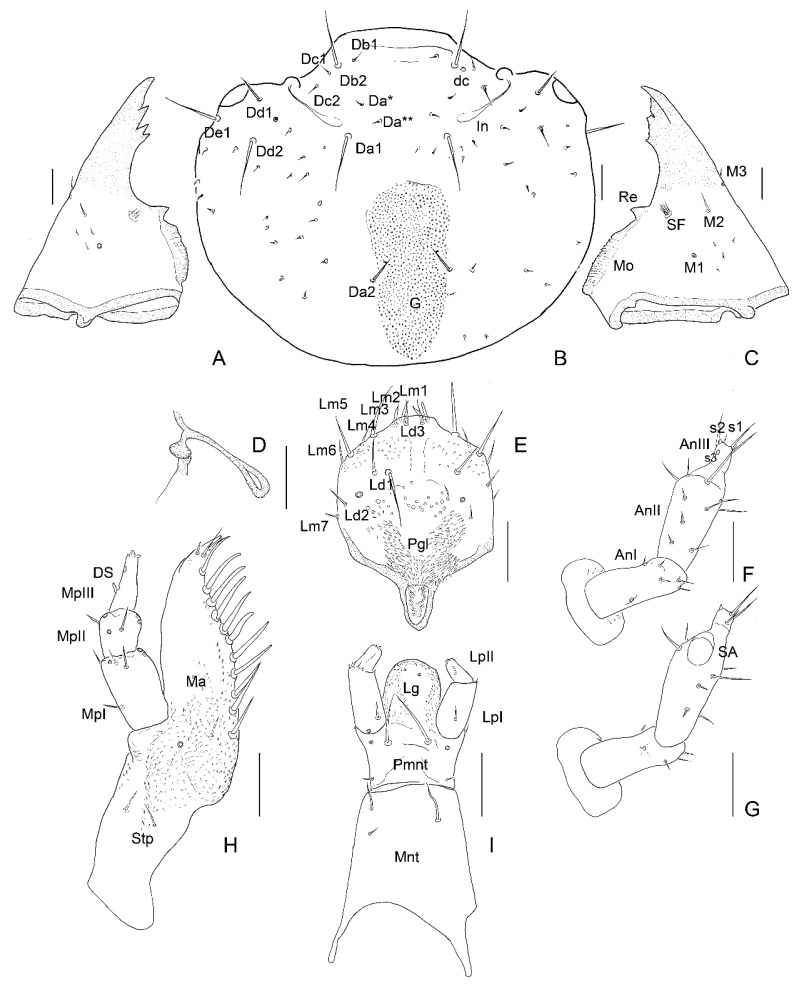
*Leiodes cinnamomea*. Details of larval morphology of instar II. (**A**) Left mandible, ventral view. (**B**) Dorsal side of head. (**C**) Right mandible, ventral view. (**D**) Sclerotized invagination of head. (**E**) Labrum, ventral view. (**F**) Antenna, dorsal side. (**G**) Antenna, ventral side. (**H**) Maxilla, ventral view. (**I**) Labium, ventral view. Abbreviations: AnI, AnII, AnIII—antennomere I, II, III; SA—sensory appendage: Cdo—cardo; DS—digitiform sensillum; G—granulation; In—invagination; LpI, LpII—labial palpi segment I, II; Lg—ligula; Ma—mala; Mnt—mentum; Mo—mola; MpI, MpII, MpIII—maxillary palpi segment I, II, III; Pgl—paraglossae; Pmnt—prementum; Re—retinaculum; SF—setal field; Stp—stipes. The other abbreviations refer to the terminology of setae. Scale bars = 0.1 mm.

**Figure 6 insects-13-00249-f006:**
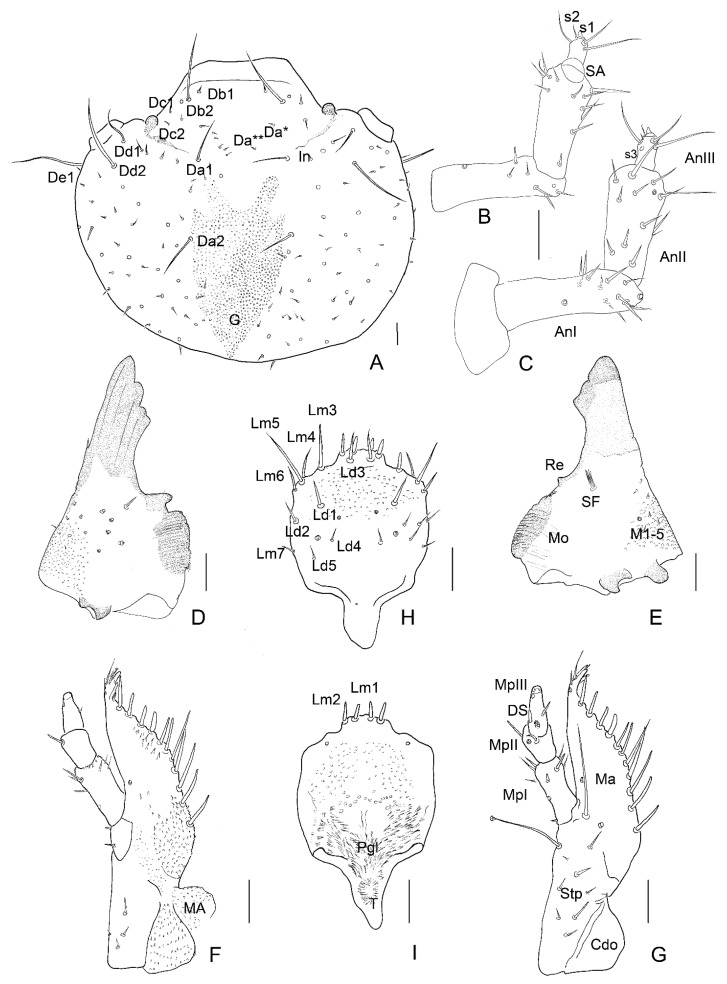
*Leiodes cinnamomea*. Details of larval morphology of instar III. (**A**) Dorsal side of head. (**B**) Antenna, ventral side. (C) Antenna, dorsal side. (**D**) Left mandible, ventral view. (**E**) Right mandible, ventral view. (**F**) Maxilla, ventral view. (**G**) Maxilla, dorsal view (**H**) Labrum, dorsal view. (**I**) Labrum, ventral view. Abbreviations: AnI, AnII, AnIII—antennomere I, II, III; SA—sensory appendage: Cdo—cardo; DS—digitiform sensillum; G—granulation, In—invagination; LpI, Lp II—labial palpi segment I, II; Lg—ligula; MA—maxillary area; Ma—mala; Mnt—mentum; Mo—mola; MpI, MpII, MpIII—maxillary palpi segment I, II, III; Pgl—paraglossae; Pmnt—prementum; Re—retinaculum; SF—setal field; Stp—stipes. The other abbreviations refer to the terminology of setae. Scale bars = 0.1 mm.

**Figure 7 insects-13-00249-f007:**
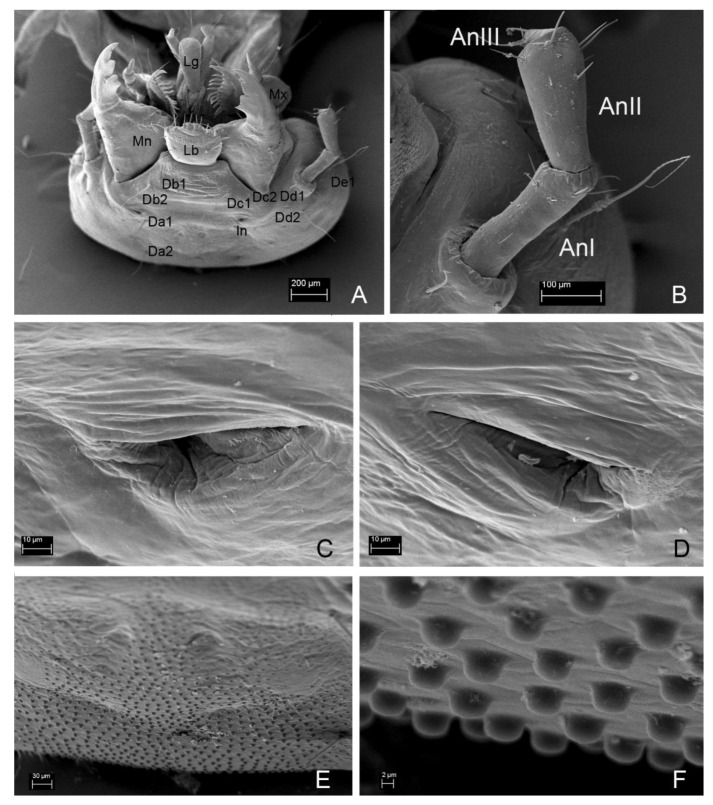
*Leiodes cinnamomea*. Details of larval morphology of instar III. (**A**) Head and mouthparts, dorso-apical view. (**B**) Antenna, dorsal side. (**C**,**D**) Invagination—pseudomycangia. (**E**,**F**) Granulation. Abbreviations: AnI, AnII, AnIII—antennomere I, II, III; L—labium; Lb—labrum; Lg—ligula; Mn—mandible; Lb—labrum. The other abbreviations refer to the terminology of setae.

**Figure 8 insects-13-00249-f008:**
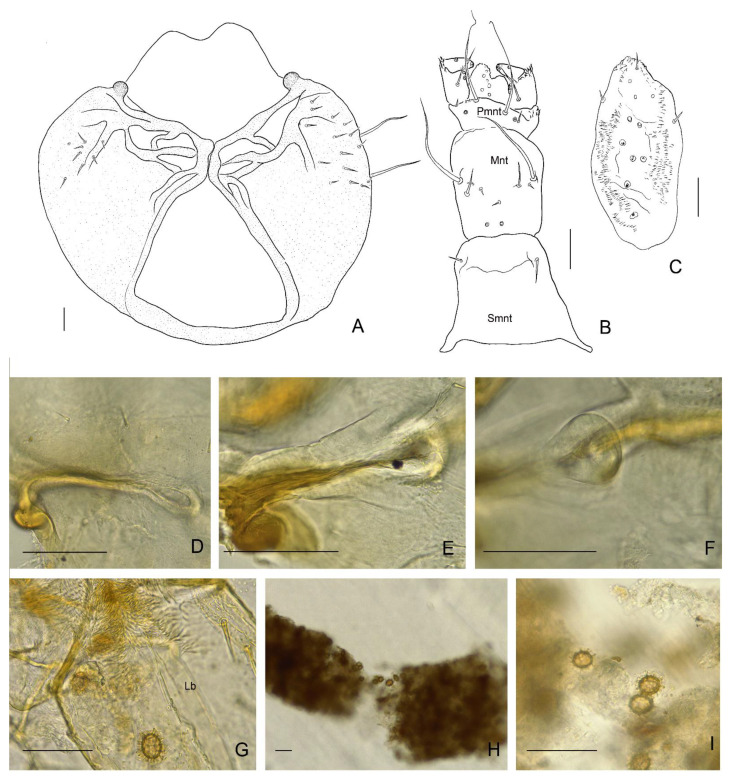
*Leiodes cinnamomea*. Details of larval morphology of instar III. (**A**) Head, ventral view. (**B**) Labium, dorsal side. (**C**) Epipharynx. (**D**–**F**) Head, invagination-pseudomycangia. (**G**–**I**) spores of *Tuber aestivum*; (**G**) on way to alimentary canal—labrum (**H**,**I**) in alimentary canal. Abbreviations: Lb—labrum; Mnt—mentum; Pmnt—prementum; Sbmnt—submentum. Scale bars for A–I = 0.1 mm.

**Figure 9 insects-13-00249-f009:**
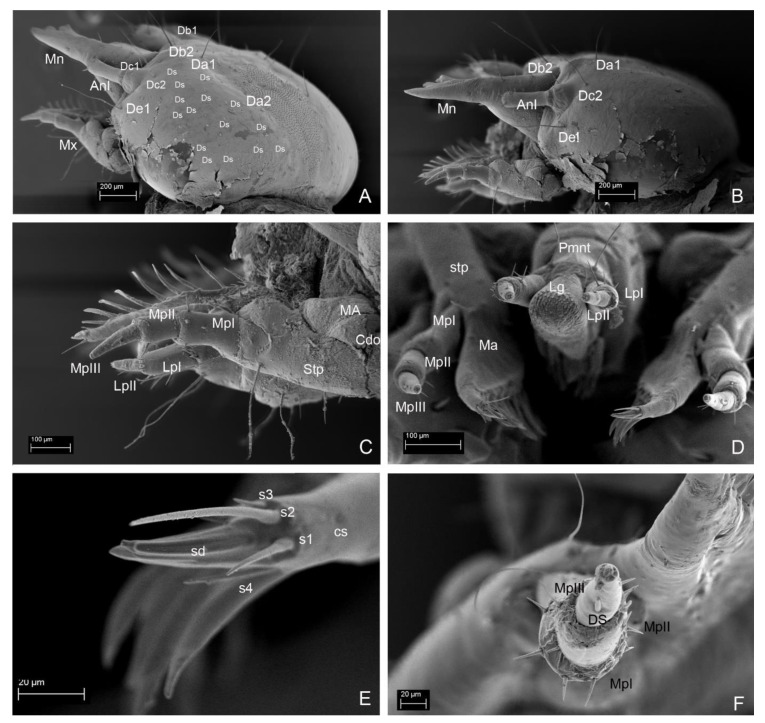
*Leiodes cinnamomea*. Details of larval morphology of instar III. (**A**) Head in dorso-lateral view. (**B**) Head in lateral view. (**C**) Maxilla, lateral view. (**D**) Labium and maxillae, apical view. (**E**) Maxilla: apex of mala. (**F**) Maxillary palp. Abbreviations: AnI, AnII, AnIII—antennomere I, II, III; Cdo—cardo; cs—campaniform sensillum; Ds—dorsal seta; DS—digitiform sensillum of maxillary palp; In—invagination; L—labium; Lg—ligula; Ma—mala; Mn—mandible; MA—maxillary area; MpI, MpII, MpIII—maxillary palpi segment I, II, III; Mx—maxilla; Pmnt—prementum; S1, 2, 3, 4—sensilla of mala; sd—sensillum digitiform of mala; Stp—stipes. The other abbreviations refer to the terminology of setae.

*Antenna*. *Instar I*. (Figure 4B). Length of antennomeres AI + AII + AIII = 0.29 mm; AI/AII/AIII/SA = 3.3/3.7/1.3/1. Antennomere I: asetose, three campaniform sensilla. Antennomere II: three setae, one large solenidium (sensory appendage SA). Antennomere III: three sensilla: two on top (process and solenidium) and one subapically; three setae. *Instar II*. (Figure 5F,G). Length of antennomeres AI + AII + AIII = 0.4 mm; AI/AII/AIII/SA = 3.9/4.7/1/1. Antennomere I: dorsal side with five setae and one campaniform sensillum, ventral side with two setae. Antennomere II: dorsal side with eight setae; ventral side with five setae and one large solenidium (SA). Antennomere III: same as in instar I, additional small seta. *Instar III*. (Figure 6B,C and Figure 7A,B). Length of antennomeres AI + AII + AIII = 0.55 mm; AI/AII/AIII/SA = 4.8/3.8/1.6. Antennomere I: dorsal side with six setae, two campaniform sensilla; ventral side with six setae and two campaniform sensilla. Antennomere II: dorsal side with ten setae and one campaniform sensillum; ventral side with 11 setae and 1 large solenidium (SA). Antennomere III: same as in instar II.

*Mandibles*. Asymmetrical. Right mola more protruding, left mola concave. Apex sclerotized, deep red-brown, proportionally to head and body large in all instars. *Instar I*: (Figure 4C,D,H). Right mandible: length: 0.48 mm, width: 0.36 mm; apex with 1 apical tooth and wide incisor area with indentation; internal edge smooth; premolar sclerotized tooth, simple; wide, short (retinaculum–re); mola (Mo) well-developed, mesially with smooth, transverse ledges (ca 18); medioventral circular and less sclerotized area (setal field SF) with ca. ten spines; three setae and two campaniform sensilla dorsolaterally. Left mandible: length 0.50 mm; width: 0.29 mm; apex with four teeth; retinaculum simple; group of trichia below mesial transverse ledges; parallel transverse rows of minute ledges dorsally. *Instar II*: (Figure 5A,C). Right mandible: length 0.60 mm, width: 0.36 mm; three additional minute setae. Left mandible: length: 0.65 mm; width: 0.34 mm; three additional minute setae. *Instar III*: (Figure 6D,E, Figure 7A, Figure 10A–F and Figure 11A,B). Right mandible: length: 0.73 mm; width: 0.56 mm; apex wide, incisor area with indentation; mesal molar surface with transverse, smooth ledges (ca 15); retinaculum simple, sclerotized, wide, short, pointed; setal field (SF) with five spines; parallel transverse rows of minute asperities dorsally; additional minute setae and granulation of epicranial area dorsally. Left mandible length: 0.88 mm; width: 0.48 mm; apex with four teeth; mesial molar surface with transverse ledges concave, sunken, more numerous than on right mandible and surrounded by trichia; setal field with two spines; parallel transverse rows of microtrichia dorsally, additional minute setae and granulation dorsally.

**Figure 10 insects-13-00249-f010:**
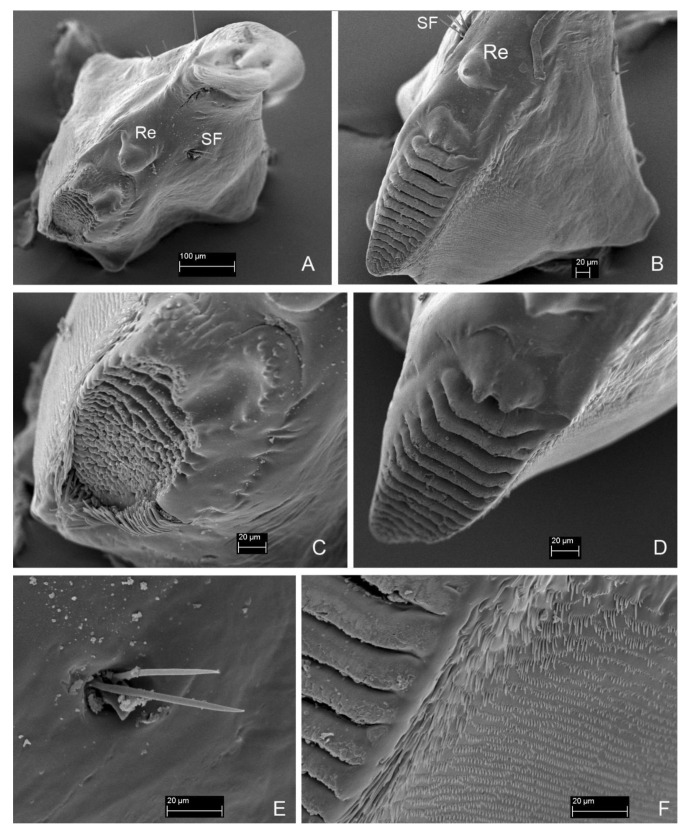
*Leiodes cinnamomea*. Details of larval morphology of instar III. Mandible. (**A**) Left. (**B**) Right. (**C**) Left, mola and retinaculum. (**D**) Right, mola and retinaculum. (**E**) Setal field. (**F**) Rigth, details of transverse ledges and rows of microtichia. Abbreviations: Mo—mola; Re—retinaculum; SF—setal field.

**Figure 11 insects-13-00249-f011:**
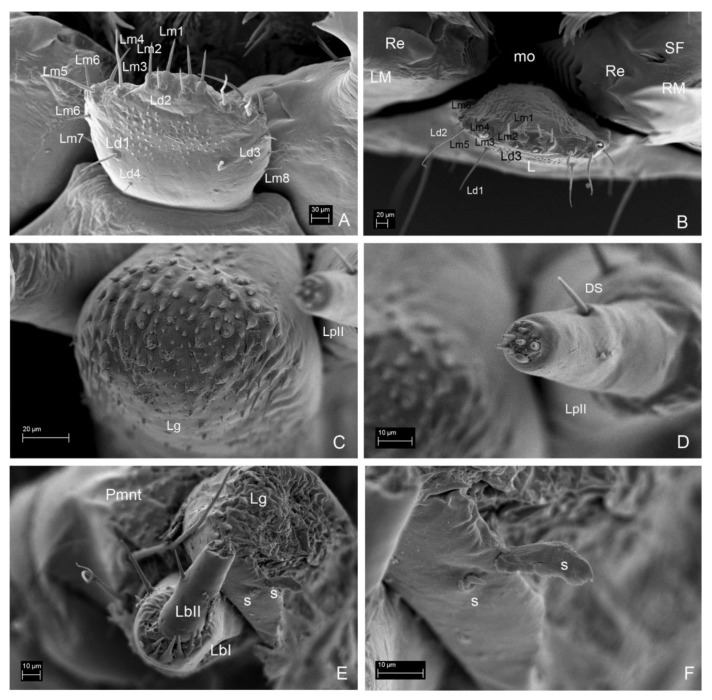
*Leiodes cinnamomea*. Details of larval morphology of instar III. Mouthparts. (**A**) labrum, dorsal side. (**B**) Labrum, apical view. (**C**) Labium—ligula, apical view. (**D**) Labial palp II. (**E**) Labial palpi and ligula. (**F**) Sensilla of ligula. Abbreviations: DS—digitiform sensillum of maxillary palp; LpI, LpII—labial palpi segment I, II; Lg—ligula; LM—left mandible; mo—mola; Mnt—mentum; Pmnt—prementum, Re—retinaculum; RM—right mandible; SF—setal field. The other abbreviations refer to the terminology of setae.

*Maxilla*. *Instar I*. (Figure 4E). MPI/MPII/MPIII = 2.5/1/2.3; Cardo (Cdo) without setae. Stipes (Stp) large with three setae. Mala (Ma) with apex undivided, with eleven large setae in row, two setae and asperities at base; apex of mala simple, with three spines laterally and one campaniform sensillum below. Maxillary palp 3-segmented; palpi segment I with two campaniform sensilla, palpi segment II with two setae and two campaniform sensilla; palpi segment III with one digitiform sensillum, one seta, one campaniform sensillum subapically and a few minute sensilla apically. *Instar II*. (Figure 5H). MPI/MPII/MPIII = 1.8/1/1.7; palpi segment I with additional two campaniform sensilla and additional three setae; palpi segment II with additional two campaniform sensilla. *Instar III*. (Figure 6F,G and Figure 9A–F). MPI/MPII/MPIII = 2/1/1.4; apex of mala with four spine-like long sensilla and one bigger digitiform sensillum (Figure 9E). Maxillary articulating area (MA) present, well-developed. Palpiger present.

*Labrum*. Longer than wide, protruding apically, tormae fused. *Instar I*. (Figure 4F). Labrum width: 0.10 mm; length: 0.13 mm. Chaetotaxy: dorsally two pairs of setae (Ld1, Ld2) and a pair of campaniform sensilla (c.s.); margin with six pairs of setae (Lm1–Lm6). Epipharynx with microtrichia posteriorly forming paraglossae; group of few sensilla. *Instar II*. (Figure 5E). Labrum width: 0.19 mm; length: 0.22 mm. Chaetotaxy: similar to instar I, additional pairs of margin seta Lm7, dorsal side with additional seta Ld3 and granulation. Epipharynx with group of numerous sensilla. *Instar III*. (Figure 6H,I, Figure 7A, Figure 8G and Figure 11A,B). Labrum width: 0.32 mm; length: 0.37 mm. Chaetotaxy: dorsally four pairs of setae (Ld1–Ld4) and a pair of campaniform sensilla (c.s.); margin with eight pairs of setae (Lm1–Lm8), dorsal and ventral side with asperities. Epipharynx with microtrichia forming paraglossae; transverse row of ca. 12 sensilla.

*Labium*. *Instar I*. (Figure 4G). Ligula simple, granulated. Prementum with pair of setae and campaniform sensilla. Mentum asetose. Labial palp: LPI/LPII = 1.74; segment I with 1 min seta latero-basally and microtrichia latero-basally and latero-apically; segment II with group of six apical peg-like sensilla and one digitiform sensillum subapically. *Instar II*. (Figure 5I) Prementum with additional pair of minute setae. Mentum with two pairs of setae. LPI/LPII = 1.84. *Instar III*. (Figure 7A, Figure 8B,C, Figure 9D and Figure 11C–F). Ligula with asperities ventrally, with a pair of minute setae and a pair of flat sensilla latero-subapically (Figure 11F). Mentum with additional two pairs of minute setae. Submentum with pair of setae. LPI/LPII = 1.6.

*Legs*. 5-segmented. Tarsungulus strongly sclerotized, brown in all instars.

All legs with granulations on dorsal side of coxa. *Instar I*. (Figure 12A–C,E). Proleg. Coxa: four setae anteriorly; six setae posteriorly. Trochanter: four setae and five campaniform sensilla anteriorly; three setae and two campaniform sensilla posteriorly. Femur: three setae anteriorly; one long seta ventrally; one seta posteriorly. Tibia: five setae anteriorly; three setae posteriorly. Tarsungulus: pair of setae. Midleg. Coxa: ten setae anteriorly; six setae posteriorly. Trochanter: four setae and five campaniform sensilla anteriorly; three setae and two campaniform sensilla posteriorly. Femur: short, three setae anteriorly; one long seta ventrally; two setae posteriorly. Tibia: six setae anteriorly; two setae posteriorly. Tarsungulus: pair of setae. Hindleg. Coxa: nine setae anteriorly; nine setae posteriorly. Trochanter: three setae and five campaniform anteriorly; five setae and two campaniform sensilla posteriorly. Femur: two setae anteriorly; two setae ventrally; one seta posteriorly. Tibia: four setae anteriorly; three setae posteriorly. Tarsungulus: pair of setae. *Instar II*. Proleg (Figure 19B). Coxa: seven setae anteriorly; eight setae posteriorly. Trochanter: five setae and five campaniform sensilla anteriorly; six setae and two campaniform sensilla posteriorly. Femur: five setae anteriorly; four setae dorsally; one seta ventrally; three setae and one campaniform sensillum posteriorly. Tibia: seven setae and campaniform sensillum anteriorly; four setae posteriorly. Tarsungulus: pairs of setae. Midleg (Figure 19C). Coxa: five setae anteriorly; six setae posteriorly. Trochanter: eight setae and five campaniform sensilla anteriorly; three setae and two campaniform sensilla posteriorly. Femur: short, four setae anteriorly; three setae posteriorly. Tibia: four setae and campaniform sensillum anteriorly; four setae posteriorly. Tarsungulus: pair of setae. Hindleg (Figure 19G). Coxa: five setae anteriorly; nine setae posteriorly. Trochanter: two setae and two campaniform sensilla; two setae and five campaniform sensilla. Femur: three setae anteriorly; four setae and one campaniform sensillum posteriorly. Tibia: five setae anteriorly; five setae posteriorly; two campaniform sensilla dorsally. Tarsungulus: pair of setae. *Instar III.* All legs with short rows of granulations. Proleg (Figure 17A and Figure 19A,D). Coxa: 17 setae anteriorly; 6 setae posteriorly. Trochanter: triangular, with nine setae and five campaniform sensilla anteriorly; six setae and two campaniform sensilla posteriorly. Femur: 14 setae anteriorly; 2 long setae ventrally and 9 setae dorsally and 15 setae posteriorly. Tibia: as long as femur; 13 setae anteriorly and 18 setae dorsally and posteriorly. Tarsungulus: with two setae and well sclerotized, brown apex. Midleg (Figure 17A and Figure 19E,F). Coxa. nine setae anteriorly; nine setae posteriorly. Trochanter triangular, with eight setae and five campaniform sensilla anteriorly; seven setae and two campaniform sensilla posteriorly. Femur. 17 setae anteriorly; 2 setae ventrally and 10 setae dorsally and 16 setae posteriorly. Tibia: as long as femur; 16 setae anteriorly and 6 setae dorsally and 23 setae posteriorly, 1 campaniform sensillum between distal setae. Tarsungulus: well sclerotized, brown apex, with two setae. Hindleg (Figure 17B and Figure 19H,I). Coxa: 17 setae anteriorly; 6 setae posteriorly. Trochanter: triangular, with 3 setae and 5 campaniform sensilla anteriorly; 13 setae and 3 campaniform sensilla posteriorly. Femur: 26 setae anteriorly; 1 long seta ventrally and 18 setae dorsally and 9 setae posteriorly, 1 campaniform sensillum between distal setae. Tibia: as long as femur; 13 setae anteriorly and 18 setae dorsally and posteriorly. Tarsungulus: well sclerotized, brown apex, with two setae.

**Figure 12 insects-13-00249-f012:**
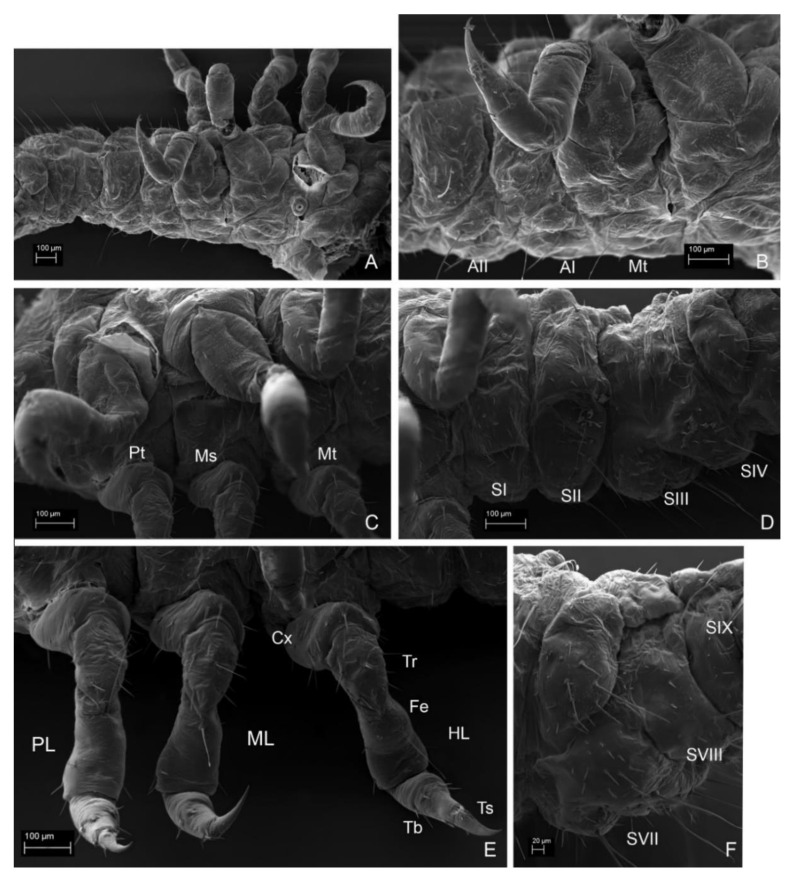
*Leiodes cinnamomea.* Details of larval morphology of instar I. (**A**,**B**) Ventro-lateral side of thorax and part of abdomen. (**C**,**E**) Legs. (**D**,**F**) Abdomen. Abbreviations: AI, II—abdominal segment I, II; Cx—coxa; Fe—femur; HL—hindleg; ML—midleg; Ms—mesothorax; Mt—metathorax; PL—proleg; Pt—prothorax; SI—IX—sternites I–IX; Tb—tibia; Tr—trochanter; Ts—tarsungulus.

*Prothorax*. Pronotum transverse. *Instar I*. (Figure 12A). Pleurites with three on precoxale, two setae in epimeron, one setae on postcoxale. Basisternum with ca. twelve setae. Sternellum with ca. 10 setae.

N1W = 0.75 mm; N1L = 0.35 mm; N1L/W = 0.47. Chaetotaxy: apically five setae Da1, Db1, Dc1, Dd1, De1; laterally seta L and three to four setae lateroventrally; posteriorly two long setae (P1, P2) and two to three small setae between; four campaniform sensilla (ds1, ds2, ds3, ps1); reservoir (R) and tergal gland (G); smooth integument. *Instar II*. (Figure 13B,D). N1W = 1.28 mm; N1L = 0.48 mm; N1L/W = 0.38. Chaetotaxy as in instar I. *Instar III*. (Figure 14A,E and Figure 17A). N1W = 1.92 mm; N1L = 0.64 mm; N1L/W = 0.33. Chaetotaxy as in instar I and II; dense asperities apicomedially. Pleurites with three on precoxale, two setae in epimeron, one seta on postcoxale. Basisternum with ca. 12 setae. Sternellum with ca. 10 setae.

**Figure 13 insects-13-00249-f013:**
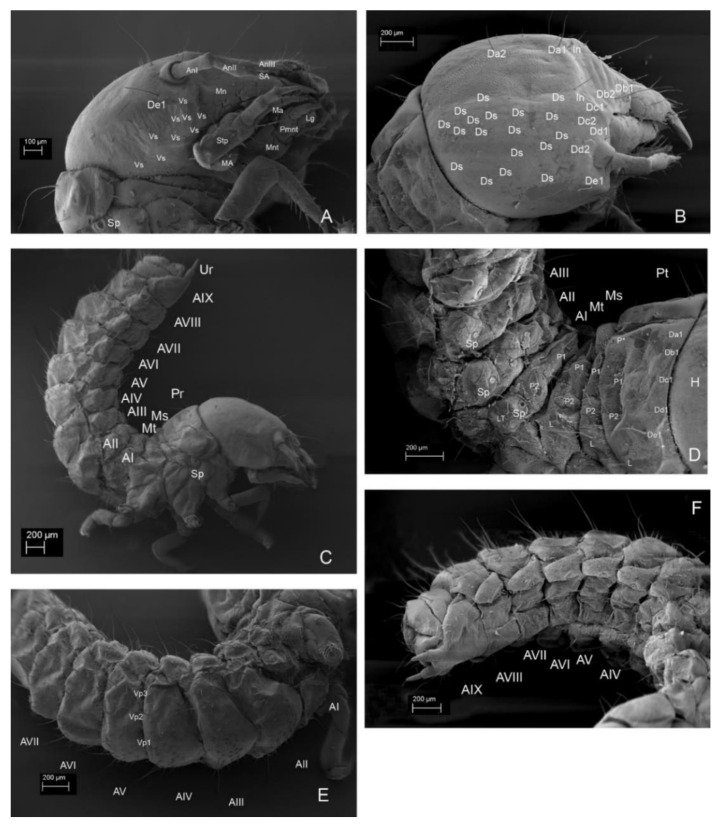
*Leiodes cinnamomea*. Details of larval morphology of instar II. (**A**) Head, ventro-lateral view. (**B**) Head, dorso-lateral view. (**C**) Body in lateral view. (**D**) Thorax and part of abdomen in dorso-lateral view. (**E**) Abdomen in ventral view. (**F**) Abdomen in dorso-lateral view. Abbreviations: AI–IX—abdominal segment I–IX; AM—anal membrane, AnI, AnII, AnIII—antennomere I, II, III; Ds—dorsal seta; G—granulation, Lb—labrum; Lg—ligula; LT—laterotergite; MA—maxillary area; Mn—mandible; Ms—mesothorax; Mt—metathorax; Mx—maxilla; Pt—prothorax; SA—sensory appendix; Sp—spiracle; Stp—stipes; Ur—urogomphi; Pmnt—prementum; VS—ventral seta. The other abbreviations refer to the terminology of setae.

**Figure 14 insects-13-00249-f014:**
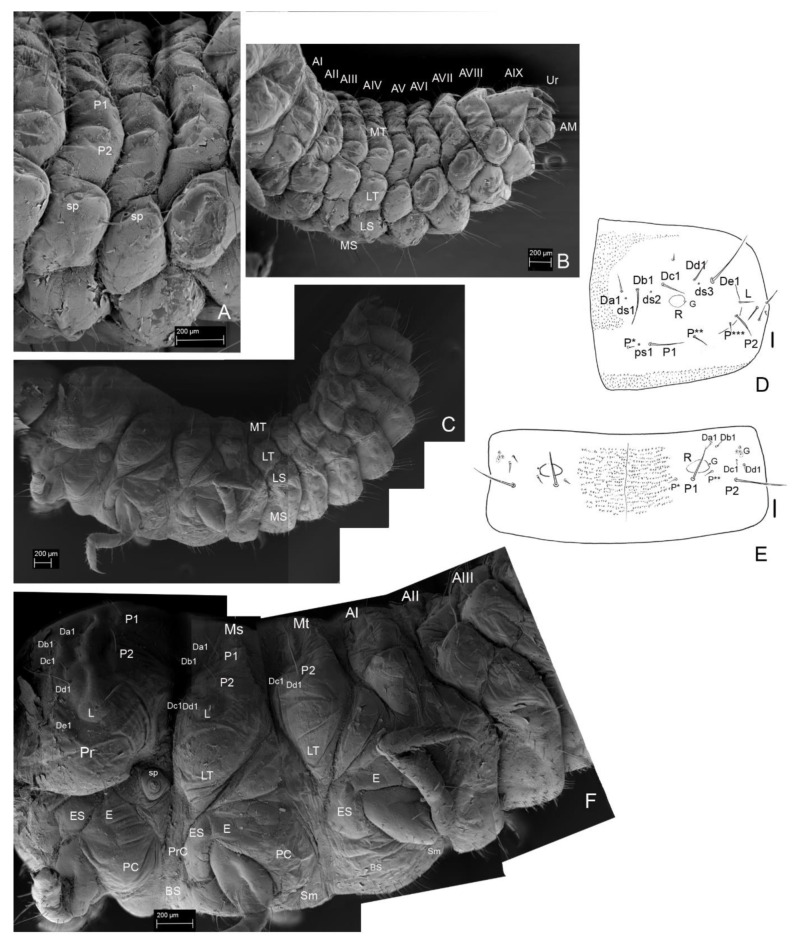
*Leiodes cinnamomea.* Details of larval morphology of instar III. (**A**) Abdominal tergites and laterotergites III–VI. (**B**) Abdominal tergites and laterotergites I–IX. (**C**) Thorax and abdomen, lateral view. (**D**) Pronotum—scheme of chaetotaxy. (**E**) Abdominal tergite—scheme of chaetotaxy. (**F**) Thorax and abdominal segments I–II, lateral view. Abbreviations: AI–IX—abdominal segments I–IX; AM—anal membrane; BS—basisternum; E—epimeron; ES—episternum; LT—laterotergite; LS—laterosternite; Mt—metathorax; MT—mediotergite; MS—mediosternite; Ms—mesothorax; PC—postcoxale; PrC—precoxale; Pt—prothorax; Sm—sternellum; Sp—spiracle. The other abbreviations refer to the terminology of setae. Scale bars for D, E = 0.1 mm.

*Mesothorax*. Mesonotum transverse. *Instar I*. (Figure 12B,C). N2W = 0.70 mm; N2L = 0.22 mm; N2L/W = 0.31. *Instar II*. (Figure 13D). N2W = 0.98 mm; N2L = 0.24 mm; N2L/W = 0.25. *Instar III*. (Figure 14D). N2W = 1.28 mm; N2L = 0.24 mm; N2L/W = 0.19. Chaetotaxy: minute Da1, Db1 anteromedially, minute setae Dc1, Dd1 anterolaterally, posteriorly long setae P1, P2 and seta (L) laterally. Metasternum: basisternum (Bs) with two to three pairs of setae and sternellum (Sm) with ca. five pairs of setae. Pleuron with two setae on precoxale and two on epimeron, two setae on postcoxalae. *Metathorax*. Metanotum transverse. (Figure 12A–C). *Instar I*. N3W = 0.67 mm; N3L = 0.16 mm; N3L/W = 0.24; chaetotaxy: minute Da1, Db1 anteromedially, minute Dc1, Dd1 anterolaterally, the longest tergal seta (L) laterally; row P with two setae P1, P2; *Instar II*. (Figure 13D). N3W = 0.80 mm; N3L = 0.24 mm; N3L/W = 0.3. *Instar III*. (Figure 14D). N3W = 0.96 mm; N3L = 0.32 mm; N3L/W = 0.33. Metasternum (basisternum (Bs) and sternellum (Sm)) with ca. 30 setae. Pleuron with two setae on precoxale and two setae on epimeron, two setae on postcoxale.

*Abdominal Mediotergites I–VIII.* Transverse, undivided, without carina. Tergites divided into mediotergites and lateral sclerites with annular spiracles. *Instar I*. (Figure 12A,B) Chaetotaxy on each side of mediotergite: medially oval reservoirs (R) with one gland (G) and seta Da1; apicolaterally two setae (Dc1, Dd1) and three glands; row P with two big setae (P1, P2), and two campaniform sensilla (ps1, ps2); Abdominal mediotergites VII–VIII with medial row of sclerotized plates: four plates in instar I. *Instar II*. (Figure 13D,F). Additional two to three small setae (P*, P**, P***) and dense microsculpture: asperities arranged into irregular short rows). *Instar III*. (Figure 14A,B,F and Figure 16C). Abdominal tergites VII–VIII with medial longitudinal row of five sclerotized plates.

*Abdominal Laterotergites I–VIII.* All latertergites with spiracle and one long seta. *Instar I.* (Figure 12B). *Instar II*. Ca. seven setae (Figure 13C–F). *Instar III*. (Figure 15A, Figure 16A and Figure 18C–F). Laterotergites I–IV with five to six setae. Laterotergites V–IX with eight to nine setae.

**Figure 15 insects-13-00249-f015:**
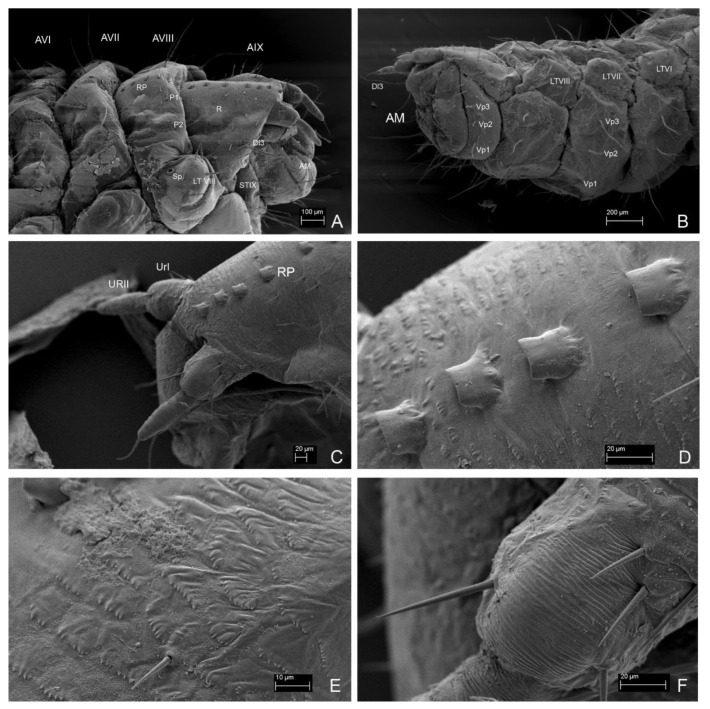
*Leiodes cinnamomea.* Details of larval morphology of instar III. (**A**) Abdominal tergites and laterotergites VI–IX and anal membrane. (**B**) Abdominal sternites and laterotergites VI–IX and anal membrane. (**C**) Abdominal tergite IX with median row of sclerotized plates and urogomphi. (**D**) Sclerotized plates of abdominal tergit IX. (**E**) Microsculpture of abdominal tergite IX. (**F**) Details of microsculpture of urogomphi. Abbreviations: AI–IX—abdominal tergites I–IX; AM—anal membrane; LTVI–IX—laterotergites VI–IX; R—reservoir; RP—rectangular plate; Sp—spiracle; UrI, II—urogomphere I, II. The other abbreviations refer to the terminology of setae.

**Figure 16 insects-13-00249-f016:**
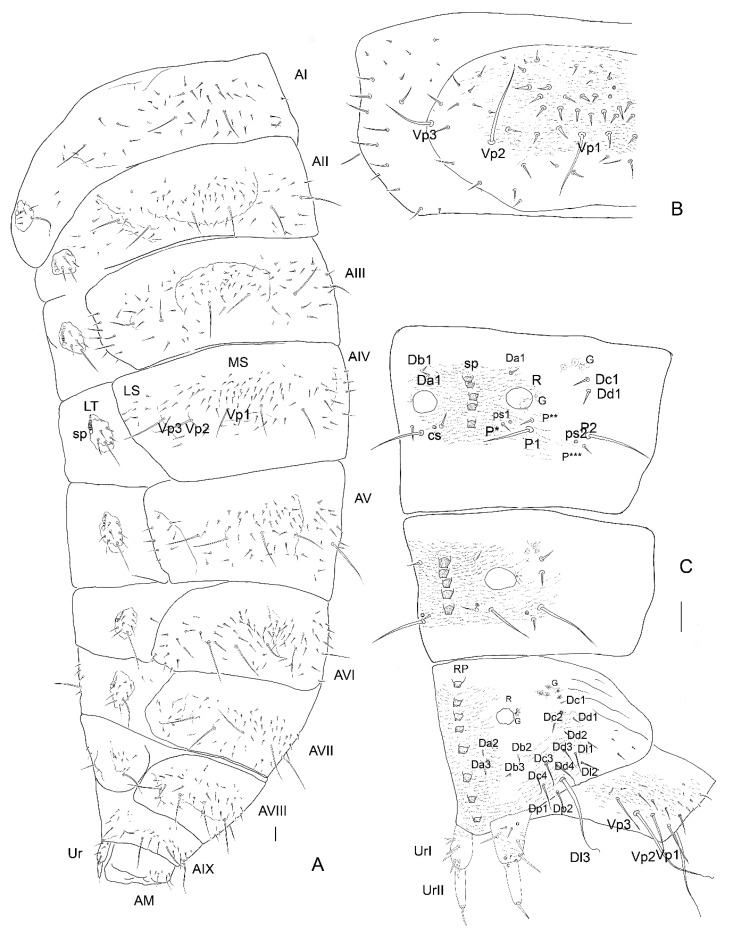
*Leiodes cinnamomea.* Details of larval morphology of instar III. (**A**) Abdominal tergites and laterotergites I–IX and anal membrane. (**B**) Abdominal sternite II. (**C**) Abdominal tergites VII–IX with urogomphi. Abbreviations: AI–IX—abdominal tergites I–IX; AM—anal membrane; LS—laterosternite; LT—laterotergite; MS—mediosternite; R—reservoir; RP—rectangular plate; Sp—spiracle; UrI, II—urogomphere I, II. The other abbreviations refer to the terminology of setae. Scale bars = 0.1 mm.

*Spiracles*. (Figure 14A,D, Figure 15A and Figure 16A). *All instars*. Annular, nine pairs of spiracles: one pair between prothorax and mesothorax and eight pairs on abdominal laterotergites I–VIII.

*Abdominal Sternites I–VIII.* Sternites with main, median plates and slightly isolated laterosternites in instar III. *Instar I*. (Figure 12A,D,F). Chaetotaxy of mediosternites: numerous short ca. 10−20 setae and 3 long postero-ventral setae (Vp1−Vp3). *Instar II*. (Figure 13E,F). Microsculpture: short, transverse rows of asperities. *Instar III*. (Figure 15B, Figure 16A,B, Figure 17B and Figure 18A–F). Same as in instar II.

**Figure 17 insects-13-00249-f017:**
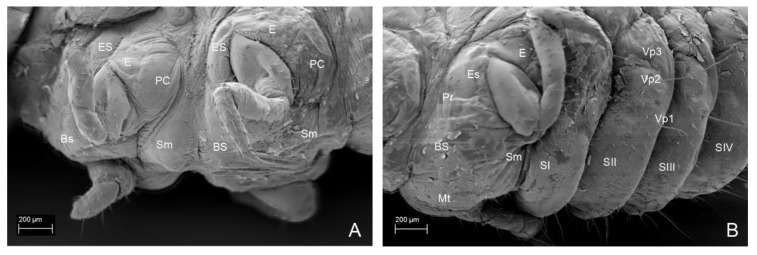
*Leiodes cinnamomea.* Details of larval morphology of instar III. Thorax in ventral view. (**A**) Pronota and mesonota. (**B**) Metanota and abdominal sternites I–III. Abbreviations: BS—basisternum; E—epimeron; ES—episternum; PC—postcoxale; Prc—precoxale; Pt—prothorax; SI–IX—abdominal sternites I–IX; Sm—sternellum; Sp—spiracle. The other abbreviations refer to the terminology of setae. Scale bars = 0.1 mm.

**Figure 18 insects-13-00249-f018:**
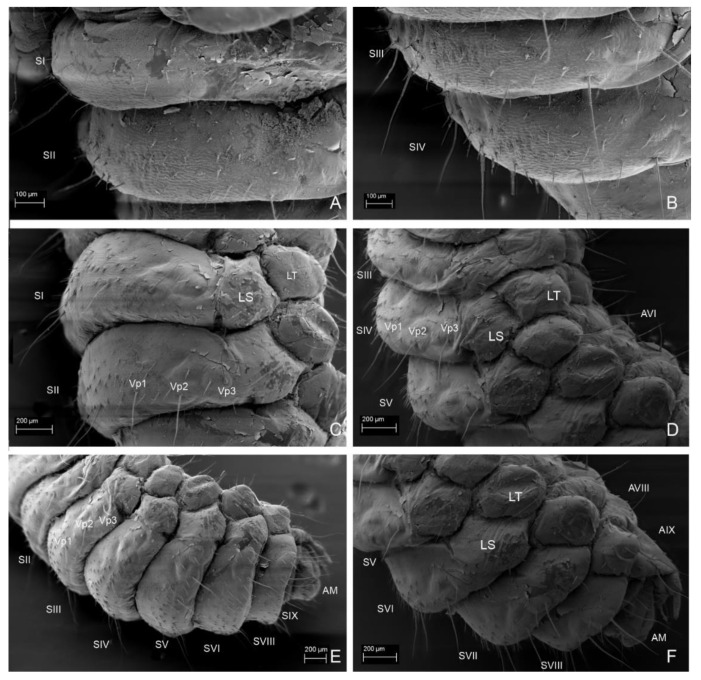
*Leiodes cinnamomea.* Details of larval morphology of instar III. Abdominal sternites. (**A**) Sternites I, II. (**B**) Sternites III, IV. (**C**) Sternites I, II, lateral view. (**D**) Sternites III–V, lateral view. (**E**) Sternites II–IX with urogomphi and anal membrane. (**F**) Sternites V–IX. Abbreviations: AI—AIX—abdominal tergites I–IX; AM—anal membrane; LS—laterosternite; LT—laterotergite; SI–IX—abdominal sternites I–IX. The other abbreviations refer to the terminology of setae.

**Figure 19 insects-13-00249-f019:**
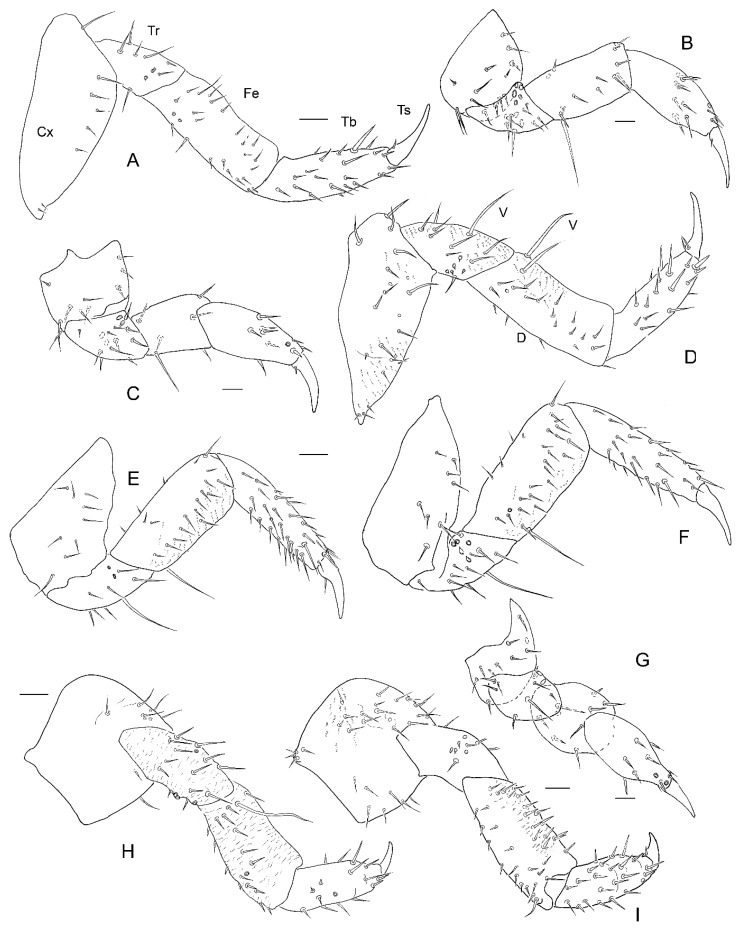
*Leiodes cinnamomea.* Details of larval morphology of legs. (**A**,**B**,**D**) Proleg. (**A**) Posterior view, instar III. (**B**) Instar II. (**D**) Anterior view, instar III. (**C**,**E**,**F**) Midleg. (**C**) Instar II. (**E**) Anterior view, instar III (**F**) Posterior view, instar III. (**G**–**I**) Hindleg. (**G**) Instar II. (**F**) Anterior view, instar III. (**I**) Posterior view, instar III. Abbreviations: Cx—coxa; Fe—femur; Tb—tibia; Tr—trochanter; Ts—tarsungulus. Scale bars = 0.1 mm.

*Abdominal Segment IX*. *Instar I*. Tergite and sternite undivided, without carina. Eight plates forming medial row. *Instar II*. (Figure 13C,F). Chaetotaxy of tergite: very large and long seta Dl3 postero-laterally (reaching even longer than to anal membrane and distal end of urogomphi). *Instar III*. (Figure 14B, Figure 15A–E and Figure 16A,C). Chaetotaxy of tergite: very large and long seta Dl3 postero-laterally; medium setae Dl2 and Dl1, 14 setae dorsally on each side of tergite (Da1–Da2, Db2–Db3, Dc1–Dc4, Dd1–Dd4, Dp1–Dp2); reservoir (R) with one opening, six glands apicolaterally on each side; short rows of asperities. Chaetotaxy of sternite: ca. seven setae lateromedially, three setae postero-ventrally (Vp1–Vp3), short rows of asperities.

*Anal membrane*. Short and wide. *Instar I*. Chaetotaxy: dorsal side with two long setae (D1); ventral side with two long setae and three medium setae. *Instar II* (Figure 13F). *Instar III*. (Figure 15A,B and Figure 16A) as in instar I: dense microtrichia; each side of ventral side with additional ca. 30 small setae.

*Urogomphi. Instar I*. Length of urogomphi: 0.2 mm. URI/URII/URS: 1.6/1.6/1. Urogomphal segment I: 4 setae, 4 campaniform sensilla. Urogomphal segment II: smooth with apical seta. *Instar II*. (Figure 13F). Length of urogomphi: 0.25 mm. URI/URII/URS: 1.4/1.8/1. Urogomphal segment I: four setae, four campaniform sensilla and transverse ledges dorsally. *Instar III*. (Figure 15A–C,F and Figure 16A,C). Length of urogomphi: 0.31 mm. URI/URII/URS: 2.1/2.1/1. Urogomphal segment I: five setae and three campaniform sensilla dorsally, transverse ledges dorsally; asperities laterally and ventrally, one seta ventrally.

## 4. Discussion

Compared to the other leiodid larvae, which are mostly campodeiform [2,32], *L. cinnamomea* has very soft bodied, grub-like, large larvae (achieving a length of 1 cm in the final instar). They have a c-inverse shaped body, with a large cranium and large, strongly sclerotized mandibles, short legs and urogomphi: all these characteristics have an adaptive role for subterranean mycophagy [35].

Unfortunately, comparison to other larvae of Leiodinae has been limited by lack of data about most of the tribes in the subfamily. Larvae of Sogdini, Estadini and Pseudoliodini were undescribed and only mentioned by Newton [36]. Descriptions of the larvae of Scotocryptini exist for three species [37,38,39]; for Leiodini, only *L. cinnamomea* has been described. The only exception is Agathidiini, where many historical and modern descriptions exist [30,31,40,41,42,43,44,45,46,47].

Head morphology of *L. cinnamomea* is worth mentioning since it possesses two unique structures for leiodid larvae: granulation (Figure 7E,F) and paired invaginations (Figure 7C,D and Figure 8D–F) on the dorsal side of the cranium. Granulation, regular, present frontally, and in all instars, seem to have a stridulatory function, discussed below. Invaginations also presented in all instars and were prolonged dorsal mandibular articulations placed symmetrically near the basement of antennae. Such cuticular pockets on the head, documented here for the first time, among leiodid larvae, are known in the adult beetle of Curculionidae (Scolytinae and Platypodinae) for transporting of symbiotic fungi [48]. Exoskelatal cavities are also distributed on other parts of the body, mainly in adult beetles, also in the leiodid Camiarinae [1,48]. They also occur on the abdominal tergites of the camiarini larvae of *Inocatops* Broun, 1893 [49]. It is believed that *L. cinnamomea* help in dispersion of spores and mycelium propagation of *Tuber* species [18,28], but functioning of these invaginations as mycangia seems to be impossible because the pockets are smaller in size than the spores (pers. obs. AK). Grebennikov and Leschen [48] suggested that such structures called mycangia-like or pseudomycangia can be reservoirs of secretions or for the thermal sensing. The last hypothesis could be an interesting explanation as to how larvae of *L. cinnamomea* can react to light without stemmata.

Comparing the morphology of the head to other larvae of Leioidinae, epicranial stem and sutures on dorsal side are absent and chaetotaxy resembles those in *Creagrophorus spinaculeus* Wheeler, 1979 [37] and *Scotocryptus meliponae* Girard, 1874 [38]. Stemmata are absent but adults have eyes present, whereas, among Leiodinae, eyes are absent in both adults and larvae in inquilines such as *S. meliponae* and *Scotocryptodes germaini* Portevin, 1907 [38,39,50]. Antennae are, proportionally to the size of the cranium, very small and short in all instars and, together with the ratio of antennomeres, are similiar to Agathidiini [30,40,41,42,43,44,45,46,47], *S. germaini* [39] and *C. spinaculeus* [37]. Sensory appendage (SA) on antennomere II is conical, short, but very wide at the base and without additional solenidia, as in *C. spinaculeus* [37] whereas other larvae of Leiodinae have two to three small sensilla (Agathidini, *S. germaini*) [30,31,39,43,44,45,46,47].

Mouthparts of the larvae of *L. cinnamomea* are characterized by large asymmetrical mandibles, with a brown-reddish apex and transverse ledges on mola (Figure 10A–D,F). They are similar to illustrations of the mandibles of larvae of *C. spinaculeus* [37]. Another leioidid larvae—*Colenisia* Fauvel, 1902 and *Nargomorphus filitarsis* Szymczakowski, 1973 [51]—have molae with ledges but are not as similar to *L. cinnamomea* as the molae of larvae of the tenebrionid bark beetle *Synchroa punctata* Newman, 1838 (Figure 36.64 in [52]), and the agyrtid larva of *Necrophilus hydrophiloides* Guérin-Méneville, 1835 [51]. This convergence of morphology of mandibular mola should point to its adaptative role to feeding on the same substrate, but, in Staphylinoidea, changing in type of food is very often not reflected in the morphology of mouthparts [51,53]. Agyrtid *Necrophilus* immatures are not sporophagous [51] but are found in small decaying carcasses and feces [36]. Leioidid *Nargomorphus* and *Creagrophorus* larvae feed on spores inside puffballs (Gasteromycetes) [36,37]. Leioidid *Colenisia* is a facultative spore feeder of Basidiomycetes [15] and guts of tenebrionid *Synchroa* larvae included woody and fungali material; fungi seem to play a leading role as nutrients [52]. Most of the spores in the alimentary canals of larvae of *L. cinnamomea* are not crushed and have visible walls and ornamentation. This poses the question ‘if undamaged spores are found in the gut, how can the beetle take nutrients from undamaged spores’?

Larvae of *Zearagytodes maculifer* (Broun, 1880) are obligate spore-feeders specialising on the bracket fungi *Ganoderma,* but ingested spores are broken in less than 10% of cases [54]. However, these ingested spores have thinner outer and inner walls and a loss of cytoplasm [55]. Larvae of *L. cinnamomea* probably cut hard material (peridium and hymenium of the fungus before it softens) using the strong apices of their mandibles; however, ledges of the mola crush spores only partially and uptaking of nutrients likely occurs in *Z. maculifer*. Although insect–fungus relationships and evolution have been investigated for decades, many questions are still unanswered and need further research.

Other parts of mouthparts—the undivided mala of the maxilla, without brushes, the protruding anterior part of labrum and the slender labium—resemble those in Scotocryptini (*S. meliponae* and *C. spinaculeus*) [37,38] and Pseudoliodini (*Colenisia*) [51]. However, there are a pair of unique sensilla on the ligula of maxilla (Figure 11F), where other leiodid larvae only have setae.

Soft-bodied larvae can have abdominal segments transversely or longitudinaly divided into folds [56]. Larvae of all instars of *L. cinnamomea* have each segment of the abdomen divided into a main tergite (mediotergite) and laterotergite, which is the area around the spiracle. Additionally, the sternites in mature larvae also look divided, forming additional lateral folds named here as laterosternites (LS) and visible on SEM photos, although they are not really separated sclerites. Chaetotaxy of the thorax and abdomen are very simple, excluding smaller setae: two pairs of long posterior setae on tergite and three pairs of long posterior setae on sternite, long seta on laterotergit near annular spiracle; this pattern is present in all instars. Only the number of smaller setae on the head, antennae, sternites and legs is different. Setae are simple-pointed apically, similar to some of Agathidiini but different to Scotocryptini, where the setae have an expanded top and are short and robust. Chaetotaxy of the abdominal sternite and the head have the same pattern as *S. meliponae* [38]. Tergites of the last three segments possess a median row of large, blunt-ended, strongly sclerotized rectangular plates (RP) on the last three tergites of the abdomen (Figure 15A,C,D): in addition to the two unique structures on the head, this is the third unique character of *L. cinnamomea* among larvae of Leiodidae. These structures were illustrated by Arzone [19] and were called *serie di scagliette rettangolari rossicce*; such structures were also described by Lyszkowski [25] for *L. rufipennis*, although he did not illustrate them. Along with granulation on the head, they seemed to be involved with very interesting behaviour, i.e., arching up of the abdomen over the heads and moving the mandibles. This most characteristic behaviour of larvae of *L. cinnamomea* was observed by Laboulbéne [26], Arzone [19] and Lyszkowski [25] for *L. rufipennis*. Lyszkowski hypothesized that these larvae could stridulate using these structures. Strong, regular, dense granulation of the head could be a *pars stridens*; the row of plates on abdomen could be a *plectra* [57]. Stridulation can be used both for calling and mating songs, but also as a defense sound, which, together with the higher posture achieved by arching and moving large mandibles, may serve as a warning signal. Unfortunately, we did not observe or record any act of stridulation. Arzone [19] stated that larvae arched their abdomen dependent on the intensity of stimulation (vibration, light), which was observed by AK, and MK; however, we did not observe that they could arch the body and bring the head close enough to the abdomen to produce sounds. However, there were no other explanations of the function of these unique structures.

During this behaviour, larvae have an inverse c-shape curvature of the body. Such positioning of the body was described for living subterranean Leiodinae by Newton [15], Wheeler [38], Baranowski [13] and, generally, as behaviour of subterranean mycophagists among beetles by Lawrence [35]. Most authors have interpreted it as a defensive behaviour, together with moving mandibles. Larvae probably protect themselves from other inhabitants of truffles, namely nematodes, acari and dipteran larvae [19,24,26,34,58] (pers. obs. AK, MK), but there are no reports of behaviour to suggest whether they are enemies of these beetles or incidental cohabitants of fungi. We observed Acarina *Zerconopsis remiger* (Figure 3A) on the truffle body but no interactions between them and the larvae. Nematodes on (Figure 3B) and inside truffle bodies were also very numerous (also recorded by Arzone [19]) as cohabitants of *L. cinnamomea*, but only one specimen was found inside the cranium of the dead larva (pers. obs. AK). Thus, it is most likely that larvae of *L. cinnamomea* defend against other individuals and their strong mandibles. However, neither Arzone [18,19] nor we observed aggregations of larvae. On the other hand, Baranowski [13] mentioned groups of three to eight larvae of *Leiodes obesa* (Schmidt, 1841) in small chambers about 4–20 cm underground. However, neither Arzone [18,19] nor we observed such aggregations of larvae. Our knowledge of the biology of subterranean fungi feeders and factors to which they are exposed is still scarce. Mycophages very often feed on ephemeral, unpredictable (spatially and temporally) sources of food [15,54,59]. Therefore, reproduction of mycophagists is often fast and is connected with a high population density (adults and immatures). Although we did not observe a high density of immatures of *L. cinnamomea* in the laboratory, we expect that, since this species is known to cause a lot of damage in economically valuable truffles, in some circumstances, larvae can be numerous and such defensive behaviour can be very important. Until now, the observations of Arzone [18,19] and ours (AK) have shown that adults lay only a few single eggs on different occasions (gradual egg laying), so larvae of different stages coexist.

It is also known that the duration and texture of fungi influence the fauna that inhabit them, influencing their biology, behaviour [59] and morphology, for both adult and larval [35,51,53]. Antipredator behaviours correlate with structure, durability and the place on the fungi, where beetles live, whether inside or on the surface [59]. Tough and persistent fungal structures correlate with the presence of egg clumping, egg insertion, aposematism, gregariousness, pupation within fungi and pupal aggregation [59]. Beetles living on, or in, ephemeral and soft fungi do not show aposematism, gregariousness, egg insertion, pupation within fungi nor pupal aggregation [59]. The most conspicuous antipredator behaviour among Leiodidae is present in the camiarinae larvae of *Z. maculifer,* which live in dense populations on wood-decaying bracket fungi (tough and persistent) *Ganoderma*, where larvae have aposematic pigmentation of the body and extremely long urogomphi [54,60]. Larval morphology of *L. cinnamomea* is characteristic of those living within fungi: its larvae are white and soft, grub-like in the last instar, C-inverse shape, with shortened legs and urogomphi, and enlarged head and mandibles. They drill corridors inside truffles and feed hymenium with spores. These features were identified by Lawrence [35] for internal feeders. However, enlarged mandibles have a reddish-brown apex, which, along with the white head, gives aposematic effect. Aposematism is a strategy for surface, persistent and tough fungi. Pupation of this species shows a mixed strategy for tough and ephemerid fungi; it can proceed inside fungi, near their surface (Figure 2D) (pers. Obs. AK, MK) or in soil, but always in cocoons (Figure 2E,F) [19] (pers. obs. AK).

To summarise, it seems that *L. cinnamomea* uses a mosaic strategy for beetles living in ephemeral and in solid fungi. Truffles, similar to other hypogean fungi, have characteristics of both of them. These fungi are quite solid and tough until maturation, when the hymenium undergoes autodigestion [35]. Are they persistent or ephemeral? Certainly, they are temporally and spatially unpredictable.

## 5. Conclusions

Soft-bodied, grub-like larva, with large head and mandibles, lack of stemmata, shortened legs, very small 2-segmented urogomphi and arching behaviour—adaptation to subterranean mycophagy.Three unique structures among leiodid larvae: exoskeletal invaginations (pseudomycangia) and granulation on head, median row of sclerotized plates on tergites VII–IX.Granulation and sclerotized plates seem to be correlated with arching up behaviour as organ of stridulation, although the act was not observed.Lack of characters of “leptinid association” of other leiodid larvae [61]: epicranial lines of head, mandibular molae with denticles, fimbriate galea.Morphology of larvae, mostly head, mouthparts, tergites and ratios of antennae and urogomphi, similar to those in Agathidiini and Scotocryptini, roughly confirmed the phylogenetic relationship in the family because we do not know larval morphology of other tribes of Leiodinae.Small differences of chaetotaxy among instars but tendency toward grub-like form of body with tergites and sternites divided into folds.Mosaic strategy—for beetles living in ephemeral and in solid fungi.

## Figures and Tables

**Table 1 insects-13-00249-t001:** Measurements and ratios of three instars of *Leiodes cinnamomea*.

	Instar I	Instar II	Instar III
total length (mm)	4.5	6.15	10
HW/HL	1.15	1.22	1.34
HW (mm)	0.81–0.88 (av. 0.84)	1.25–1.42 (av. 1.34)	1.6
HL (mm)	0.69–0.78 (av. 0.73)	0.90–1.28 (av. 1.1)	1.00–1.44 (av. 1.19)
N1W (mm)	0.75	1.28	1.92
N2W (mm)	0.7	0.98	1.28
N3W (mm)	0.67	0.8	0.96
A1W (mm)	0.51	1.2	1.76
N1L (mm)	0.35	0.48	0.64
N2L (mm)	0.22	0.24	0.24
N3L (mm)	0.16	0.24	0.32
A1L (mm)	0.13	0.24	0.32
N1L/W	0.47	0.38	0.33
N2L/W	0.31	0.25	0.19
N3L/W	0.24	0.3	0.33
A1L/W	0.25	0.2	0.18
Antenna length (mm)	0.29	0.4	0.55
AI/AII/AIII/SA	3.3/3.7/1.3/1	3.9/4.7/1/1	4.8/3.8/1.6
Labrum width (mm)	0.1	0.19	0.32
Labrum length (mm)	0.13	0.22	0.37
Mandible width (mm)	L 0.29/R 0.36	L 0.34/R 0.36	L 0.48/R 0.56
Mandible length (mm)	L 0.50/R 0.48	L 0.65/R 0.60	L 0.88/R 0.73
MPI/MPII/MPIII	2.5/1/2.3	1.8/1/1.7	2/1/1.4
LPI/LPII	1.74	1.84	1.6
Urogomphal length (mm)	0.2	0.25	0.31
URI/URII/URS	1.6/1.6/1	1.4/1.8/1	2.1/2.1/1

## Data Availability

Not applicable.

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
