# Peer review of "Larval Morphological Adaptations of *Leiodes cinnamomea* (Panzer, 1793) (Coleoptera: Leiodidae: Leiodinae)—Obligatory Feeder of *Tuber* Species"

_insects, 2022, doi:10.3390/insects13030249_

Round 1

Reviewer 1 Report

The authors described morphology of three larval stages of Leiodes cinnamomea, with well-designed figure plates. As only very limited information of larval morphology is known in the group Leiodini, the manuscript provided very detailed description and could be used as an exemplar for future studies. The introduction and discussion are also very informative, which makes this article even more valuable. I recommend that it could be published after minor revision.

However, it would be helpful to provide more SEM images of larvae of the 1st and 2nd instars, especially for details such as mouthparts, antennae, etc. The line-drawings are great, but it would not harm the work if some clear SEM images with good quality are added,  as it is mentioned in the Materials and methods that all three instars have already been investigated under SEM.

Minor issues related to format or grammar:

Line 241: Should A1 be AI?

Line 244: AI/AII/AIII/AS=4.8/3.8/1.6/? and AS=SA (sensory appendage)?

Fig. 4, 5 and many other plates: the abbreviations of setae of the larvae (eg. Dc, Dd, De etc. ) should be mentioned in the legends.

Fig. 8 D-I: scale bars are missing

Fig. 13: labels in black are not clearly visible

Fig. 16 and 18: the word “sternit” (singular form) should be replaced by “sternite”

Author Response

Dear Reviewer,

Thank you for your time and all comments, which approved the manuscript.

Our replies on your recommendation or remarks are given below.

However, it would be helpful to provide more SEM images of larvae of the 1st and 2nd instars, especially for details such as mouthparts, antennae, etc. The line-drawings are great, but it would not harm the work if some clear SEM images with good quality are added,  as it is mentioned in the Materials and methods that all three instars have already been investigated under SEM.

The first instar larva of this species has very thin and delicate cuticle so preparing mouthparts in the right way is difficult, because such membranous structures tend to collapse during dehydration (the first step of preparing to SEM). What’s more, this material was difficult to coat with gold and few attempts was made although we used the very good vacuum coater Leica EM ACE250, which allows to coat with nano-layers. We had to decide what is possible to do with limited time, material of the first instar and money to make SEM. Instar II is, in all leiodid larvae, very similar to instar III (in instar III only small, additional setae are added) that’s why, once again, it was a decision: what is more important to show. We also try to limit SEM and illustrations to reasonable number to make the study clear, easy to reading and understanding.

****

Minor issues related to format or grammar:

Line 241: Should A1 be AI?

Corrected

Line 244: AI/AII/AIII/AS=4.8/3.8/1.6/? and AS=SA (sensory appendage)?

Yes, corrected.

Fig. 4, 5 and many other plates: the abbreviations of setae of the larvae (eg. Dc, Dd, De etc. ) should be mentioned in the legends.

Please, notice that mentioning one by one all setae, lengthen the legend and make it unclear. We did not find a logic key how to do according to reviewer suggestion so we propose to add, in the legend, at the end of the text: “the other abbreviations refer to the terminology of setae”.

Fig. 8 D-I: scale bars are missing

Added.

Fig. 13: labels in black are not clearly visible

Done.

Fig. 16 and 18: the word “sternit” (singular form) should be replaced by “sternite”

Corrected.

Reviewer 2 Report

Nice contribution to our understanding of the larval morphological  adaptations of Leiodes cinnamomea to feeding. The paper is very well written and the illustrations are fantastic and informative. I would like to recommend acceptance after a very minor revision: a 

1. Title, Change Morphological larval adaptations to 'Larval morphological  adaptations'

2. Table 1.  Make it as three-line form.

Line 458: ...,43,44,45,46,47] citation format reconsidered.

Lines 550-563:  All conclusion lines should be in full sentences. 

Author Response

Dear Reviewer,

Thank you for your time and all comments, which approved the manuscript.

Our replies on your recommendation or remarks are given below.

  1. Title, Change Morphological larval adaptations to 'Larval morphological  adaptations'

Corrected.

  1. Table 1.  Make it as three-line form.

Corrected.

Line 458 ...,43,44,45,46,47] citation format reconsidered.

We stay it because we are not sure what editor prefers. In some articles is 43-47 but in some all number of references are provided, and two other reviewers and editor did not correct it.

Lines 550-563:  All conclusion lines should be in full sentences

We think that this form of conclusion is more concise and clear to summarize. Two other reviewers accepted this form.

Reviewer 3 Report

It is necessary to make some corrections:

[435] Larvae of Sogdini, Estadini and Pseudoliodini were undescribed, and only mentioned by ????[36].

[578] Acknowledgments: We are grateful to Prof. Andrzej Warchałowski (University of Wrocław) and...

"the late Professor Andrzej Warchałowski"

REFERENCES:

[601] 7. Perreau, M. Leiodidae. In Catalogue of Palaearctic Coleoptera, Vol. 2: Hydrophiloidea-Staphylinoidea. Löbl, I., Smetana, A., Eds.; 601 Stenstrup: Apollo Books. 2004, pp. 133‒203.

now is avaiable updated edition:

Löbl I. & Löbl D. (eds) 2015: Catalogue of Palaearctic Coleoptera. Hydrophiloidea-Staphylinoidea. Vol. 2. Revised and updated edition. Brill, Leiden, Boston, xxvi + 1702 pp. ISBN 978-90-04-28992-5 (print book), ISBN 978-90-04-29685-5 (e-book). 

[681] Kilian, A. Studies on the morphology of immature stages of the tribe Agathidiini (Coleoptera: Leiodidae). Part IV. Agathid-681 ium angulare Mannerheim 1852. [Annales Zoologici???] 2006, 56(4), 587–600.

Specific comments:

  1. What is the main question addressed by the research?

It only description, detailed but without any general remarks as about taxonomy, biology.

  1. Do you consider the topic original or relevant in the field, and if so, why?

Of course, this is a very important scientific achievement, original because it was unknown so far. Only less than 10% of known beetle species have the described larvae.

  1. What does it add to the subject area compared with other published material?

The description of the unknown larva is the original new data - see previous point

  1. What specific improvements could the authors consider regarding the methodology?

Comparison of larval chaetotaxy (treated as a main tool for evolutionary research and systematic) and other structure in related species with an attempt to determine homologies

  1. Are the conclusions consistent with the evidence and arguments presented and do they address the main question posed?

Yes, except the subject homology (poin 4)

  1. Are the references appropriate?

Yes, except: [601] 7. Perreau, M. Leiodidae. In Catalogue of Palaearctic Coleoptera, Vol. 2: Hydrophiloidea-Staphylinoidea. Löbl, I., Smetana, A., Eds.; 601 Stenstrup: Apollo Books. 2004, pp. 133‒203.

  1. Please include any additional comments on the tables and figures.

Well done with clasical structure.

Author Response

Dear Reviewer,

Thank you for your time and all comments, which approved the manuscript.

Our replies on your recommendation or remarks are given below.

All small corrections are done:

[435] Larvae of Sogdini, Estadini and Pseudoliodini were undescribed, and only mentioned by ????[36].

 [578] Acknowledgments: We are grateful to Prof. Andrzej Warchałowski (University of Wrocław) and...

"the late Professor Andrzej Warchałowski"

REFERENCES:

[601] 7. Perreau, M. Leiodidae. In Catalogue of Palaearctic Coleoptera, Vol. 2: Hydrophiloidea-Staphylinoidea. Löbl, I., Smetana, A., Eds.; 601 Stenstrup: Apollo Books. 2004, pp. 133‒203.

now is avaiable updated edition:

Löbl I. & Löbl D. (eds) 2015: Catalogue of Palaearctic Coleoptera. Hydrophiloidea-Staphylinoidea. Vol. 2. Revised and updated edition. Brill, Leiden, Boston, xxvi + 1702 pp. ISBN 978-90-04-28992-5 (print book), ISBN 978-90-04-29685-5 (e-book).

[681] Kilian, A. Studies on the morphology of immature stages of the tribe Agathidiini (Coleoptera: Leiodidae). Part IV. Agathid-681 ium angulare Mannerheim 1852. [Annales Zoologici???] 2006, 56(4), 587–600.

Reply on specific comments:

 Ad. 1.  What is the main question addressed by the research?

 It only description, detailed but without any general remarks as about taxonomy, biology.

Whole discussion is a general  remark on: comparative larval  morphology of the studied species and other leiodid larvae, about evolutionary, adaptative aspects its morphology and biology (behaviour, habitat, feeding, egg laying, blindness). Taxonomic position of Leiodes cinnamomea is solid and phylogenteic analyse based on larval morphology of one representative of Leiodini would be very preliminary, artificial and useless

Ad. 4.   What specific improvements could the authors consider regarding the methodology?

 Comparison of larval chaetotaxy (treated as a main tool for evolutionary research and systematic) and other structure in related species with an attempt to determine homologies

 We determined homologies of chaetotaxy and other morphological structures. Whole chaetotaxy used in this study is based on the system, which has been made and used for staphylinoid and for most of known leiodid larvae. It is written in the methodology (112):

“Terminology used for the chaetotaxy follows systems proposed by Ashe and Watrous [29], Wheeler [30] and Kilian [31,32].”

We agree that larval chaetotaxy is an important tool and the first author used this tool in previous studies but, because of arguments in the point 1, we did not use it for any phylogenetic analyses or systematic changes here. The most important comparisons, morphological differences and remarks on the placement of larva of L. cinnamomea are provided in the discussion and the conclusion.

Ad. 5.   Are the conclusions consistent with the evidence and arguments presented and do they address the main question posed?

 Yes, except the subject homology (point 4)

Please, see above.

 Ad. 6.  Are the references appropriate?

 Yes, except: [601] 7. Perreau, M. Leiodidae. In Catalogue of Palaearctic Coleoptera, Vol. 2: Hydrophiloidea-Staphylinoidea. Löbl, I., Smetana, A., Eds.; 601 Stenstrup: Apollo Books. 2004, pp. 133‒203

Corrected.